# The Preservation of the Effects of Preweaning Nutrition on Growth, Immune Competence and Metabolic Characteristics of the Developing Heifer

**DOI:** 10.3390/ani13081309

**Published:** 2023-04-11

**Authors:** Emma M. Ockenden, Victoria M. Russo, Brian J. Leury, Khageswor Giri, William J. Wales

**Affiliations:** 1Agriculture Victoria, Ellinbank, VIC 3821, Australia; 2Faculty of Science, The University of Melbourne, Parkville, VIC 3010, Australia; 3Centre for Agricultural Innovation, The University of Melbourne, Melbourne, VIC 3010, Australia; 4Agriculture Victoria, Bundoora, VIC 3083, Australia

**Keywords:** heifer development, calf nutrition, immune challenge, accelerated milk feeding, heifer calf development

## Abstract

**Simple Summary:**

This experiment is the postweaning phase of a series of experiments following 20 dairy replacement heifers from birth through to 22 months of age. During the preweaning phase (outlined in a companion paper), calves fed a high volume of milk (8 L per day) had superior growth, immune competence and metabolic characteristics when faced with a vaccination immune challenge than those on a lower volume of milk (4 L per day). Postweaning, these calves were treated equally under non-experimental conditions, and these systems were re-examined with a repeat immune challenge at 12 months of age (current experiment). Findings from this immune challenge suggest immunological imprinting from the preweaning nutrition. Superior immune responses were found in the heifers fed high volumes of milk in the preweaning phase, despite a period of 9 months of no nutritional treatment. Despite the greater immune response, there was no difference in the metabolic biomarkers detected between the two preweaning treatment groups, suggesting no trade off or compensation of energy to other processes for this superior response. Accelerated growth from the low treatment group resulted in no differences in bodyweight between the treatment groups at 13 months of age. The results indicate a positive influence for improved preweaning nutrition on heifer immune response until at least 12 months of age.

**Abstract:**

This experiment investigated the preservation effects of two preweaning milk feeding nutritional treatments (High: 8 L and Low: 4 L milk per day) on 20, 12-month-old Holstein-Friesian dairy heifers (*Bos taurus*). A vaccination immune challenge was initially implemented on these 20 heifers at 6 weeks of age and the findings indicated superior growth, immune competence and favorable metabolic characteristics from the calves that had been fed 8 L milk per day. Postweaning, all heifers were treated the same under non-experimental conditions, and the immune challenge was repeated at 12 months of age for the current experiment. Consistent with the first immune challenge, heifers from the High preweaning treatment group still had higher white cell count and neutrophil count, indicating superior immune competence. The differences found in metabolic biomarkers, including beta-hydroxybutyrate, glucose and insulin, in the preweaning phase had disappeared, suggesting these biomarkers were influenced directly by the nutritional input at the time. There were no differences in NEFA levels between treatments at either stage of development. Postweaning, the heifers from the Low preweaning treatment group experienced accelerated growth with slightly numerically higher ADG (0.83 kg/day vs. 0.89 kg/day), resulting in the initial differences in bodyweight recorded at weaning being eliminated by 13 months of age. These results are evidence of a form of immunological developmental programming as a result of accelerated preweaning nutrition and therefore, are not supportive of restricted milk feeding of calves.

## 1. Introduction

The rearing of replacement heifers is one of a dairy farmers most important investments and a leading expense on dairy farms [1]. The period from birth until first calving is the non-productive phase of a dairy cow’s life, costing Australian farmers between AUD 1500–AUD 2100 per replacement heifer (Dairy Australia, personal communication, 2022). In Victoria, between 30 and 35% of these replacement heifers die or are culled before their second calving [2]. Rearing costs are rarely met until this time, so many do not return their initial investment [1,3,4]. The resilience of an animal refers to its overall performance and relies on both immune competence and metabolic efficiency. A dairy cow with greater resilience will consistently produce high levels of milk, reproduce without difficulties, suffer few infectious or production-associated diseases and have greater longevity [5,6]. Therefore, a cow with suboptimal resilience can rapidly become unhealthy and, in turn, unprofitable [7]. The development of these physiological systems is widely understood to be influenced by early life nutrition [8]; it is therefore vital that optimal calf rearing strategies are utilised to ensure production of more resilient heifers and cows. Production of more resilient heifers would reduce the number of replacement animals required, which not only improves health, welfare and profitability but also contributes to reducing the carbon footprint of the industry [1,6,9].

Accelerated nutrition in the preweaning phase has been shown to have long term benefits on cow health and productivity through a form of developmental programming or imprinting. However, the functionality and critical periods of this development are not well understood [10,11,12]. It has been demonstrated that early life nutrition plays a major role in the future production potential of a heifer with several studies finding that accelerated preweaning nutrition results in greater milk yield in the first lactation [10,12] and also subsequent lactations [11]. Whereas programming of milk production potential is occurring, the effect that preweaning nutrition has on the future immune and metabolic function is relatively unknown. Human studies have investigated the link between early life exposures, such as nutrition, and their influence on the development of immune cells and their pathways. The development of these pathways and cells have specific critical periods that contribute to or imprint on their long-term immune susceptibility [13]. Metabolic programming of the calf begins in utero, yet little is known about the influence accelerated preweaning nutrition has on long-term metabolic health [14]. The little existing evidence that is available is conflicting and has reported accelerated preweaning nutrition as beneficial [15], futile [16,17] or even potentially detrimental depending on the prenatal environment [14]. 

This experiment details the postweaning phase of a series of experiments designed to determine strategies that lead to increased milk production and resilience of cows through manipulation of early life nutrition. In the preweaning phase, these calves were followed from birth to weaning and the effects of feeding either 4 L (Low) or 8 L (High) of milk per day on growth, immune competence and the metabolic state of calves was investigated and is reported in Ockenden et al. [18]. The current industry standard for milk feeding calves is restricted to 4 L per day; however, results from that experiment are indicative of a positive influence of accelerated preweaning nutrition. White cell count and neutrophil counts were significantly higher in the High treatment group post vaccination, suggestive of superior immune competence. Favourable metabolic characteristics were also observed in the High treatment group with significant differences in the levels of BHB, glucose and insulin recorded between nutritional treatments [18]. This current experiment was designed to determine if the differences found in the preweaning phase were preserved after a period of no nutritional treatments. This was performed by repeating the same immune challenge implemented in the preweaning phase at 12 months of age. Immune challenges are considered the gold standard when assessing immunity [7] and are scarce throughout calf nutritional studies. Further, to our knowledge they have not been utilised in repeat studies examining the effects of accelerated nutrition on post-weaned heifers. The hypothesis tested were (1) that calves from the High preweaning treatment group would continue to have greater average body weight than calves from the Low preweaning treatment group until 22 months of age and (2) that the differences observed in immune competence and metabolic characteristics in the preweaning phase would still be prevalent between preweaning treatments after a period of no nutritional treatments at 12 months of age. 

## 2. Materials and Methods

Experimental procedures were undertaken at Agriculture Victoria Dairy Research Centre’s heifer rearing contractor property in Trafalgar, Victoria Australia (38°21′ S,146°15′ E) in July–August 2021. Experimental protocols were approved by the Agricultural Research and Extension Animal Ethics Committee, Application number: 2021-06.

### 2.1. Experimental Design and Treatments

This experiment used 20 Holstein-Friesian replacement dairy heifers that were previously enrolled in a preweaning calf experiment conducted in July 2020 [18]. At birth, calves were randomly assigned to one of two nutritional treatments (4 L vs 8 L of milk daily) that was fed from birth until weaning at 10 weeks of age. Postweaning, these calves were treated as per normal farm practice, i.e., no differential nutritional treatments were applied and both treatment groups were treated the same. 

Immune challenge experimental procedures ran over a 10-day period when the heifers were approximately 12 months of age. All other data, including liveweight and reproductive performance data was collected as available commercial data. 

### 2.2. Animal Management 

#### 2.2.1. Preweaning Animal Management

Animal selection and preweaning animal management is outlined in the companion paper Ockenden et al. [18]. In summary, calves were separated from their dam within 8 h of birth and tube fed 4 L of high-quality colostrum (>22% Brix). All calves were tested and received successful passive transfer of immunity prior to enrolment in the experiment. Treatments were balanced for birth bodyweight and estimated Balance Performance Index (BPI) (Australia’s national selection index). Calves were group housed within a single pen. All calves were manually fed their daily allocation of whole milk (either 4 L or 8 L) over two equal feeds (2 L or 4 L) in the morning and at night. Concentrate and hay was available ad-libitum within the pen and individual intakes were recorded. Each calf commenced their first immune challenge at 42 days of age. Calves completed the experiment at weaning [18]. 

#### 2.2.2. Postweaning Animal Management

Experimental heifers were run within a single herd of approximately 120 other heifers of a similar age. All animals had the same access to grazed pasture as majority of the diet. If pasture became limiting (due to seasonal variability), the diet was supplemented with pasture silage fed out in the paddock with the objective of providing diet ad libitum. For the first 3 months postweaning, the diet was also supplemented with an additional commercial heifer growing concentrate (Reid Stockfeeds-Gippsland, Trafalgar, VIC, Australia) as per the normal farm practice of the Agriculture Victoria Dairy Research Centre heifer rearing facility. 

Heifers were weighed monthly using electronic scales as part of normal farm practice. This data was collected for bodyweight comparisons between the previously applied preweaning treatments.

### 2.3. Immune Challenge

Heifers underwent the same immune challenge as in Ockenden et al. [18]. These immune challenges consisted of a modified protocol of Aleri et al. [19,20,21]. A commercially available vaccine for bovine respiratory disease (BRD) (Bovilis MH + IBR; Coopers Animal Health, Macquarie Park, NSW, Australia) was used to interrupt homeostasis allowing the subsequent responses to be compared between treatments. 

Measurement protocols for this immune challenge were further refined since those previously imposed on these heifers when they were 6 weeks of age [18]. The refinement of procedures includes the removal of an interim blood test and skinfold measurements being taken. In the preweaning phase, the immune responses generated were greater on day 10 than on days 8 and 9 of the immune challenge; therefore, these interim measurements were eliminated. The remaining procedures included a total of two blood samples, taken on the first and the final (day 10) days of the immune challenge. Two skinfold measurements were taken on day 8 and repeated on day 10, which are further outlined below.

#### 2.3.1. Acquired Immune Response

On day 1 of the immune challenge, the initial blood sample was taken from each heifer prior to vaccination. The blood test was then repeated on day 10 to determine individual responses to the vaccination. Blood samples were collected from each heifer via caudal venepuncture. Each sample required 35 mL of blood and was collected into four vacutainers (lithium heparin (10 mL), plain (10 mL), EDTA (10 mL) and fluoride/oxalate (5 mL). Plain tubes were immediately placed in an incubator at 24 °C for at least 2 h prior to being centrifuged at 1258× *g* for 15 min at 24 °C. All other tubes (Lithium heparin, EDTA and fluoride/oxalate) were immediately stored on ice prior to centrifugation at 1578× *g* for 15 min at 4 °C. Additionally, a subsample of whole blood was extracted from the EDTA tube prior to centrifugation and sent to a commercial laboratory for urgent haematology analyses (Gribbles Veterinary Pathology, Clayton, VIC, Australia). All serum and plasma samples were stored at −20 °C until the analyses could be completed. 

#### 2.3.2. Cell Mediated Immune Response

Hypersensitivity skinfold testing procedures began 8 days post vaccination. As mentioned, the measurement protocols were refined from previous experiments including Aleri et al. [19,20,21] and Ockenden et al. [18]. Skinfold thickness measurements were taken 3 times using Harpenden skinfold callipers (Creative Health Products Inc., Ann Arbor, MI, USA) prior to injection as the baseline measurement. The BRD vaccine (0.1 mL) was then injected intradermally into the right caudal fold (test site) and 0.1 ml of saline was injected intradermally into the left caudal fold (control). Reactions were assessed at 48 h by comparing changes in skinfold thickness at the test site in relation to the control site. The response was corrected using the following formula from Aleri et al. [19,20,21]:
Increase (mm) = (A − B) − (C − D),(1)
where by

A = mean test site thickness at 48 h post injection;B = mean test site thickness prior to injection;C = mean control site thickness at 48 h post injection;D = mean control site thickness prior to injection.

### 2.4. Biomarker Analysis

Analyses for both samples included metabolic biomarkers, including beta-hydroxybutyrate (BHB), non-esterified fatty acid (NEFA), glucose and insulin, and immune biomarkers, including white blood cell count (WBC) and differential and infectious bovine rhinotracheitis (IBR) antibodies. 

Metabolic biomarkers assays, including those for BHB, NEFA and glucose, were performed at AgriBio labs (Bundoora, VIC, Australia) using Catachem Inc. reagents, controls and calibrators as per manufacturer’s instructions, on a ChemWell 2910 (Oxford, CT, USA) automated analyser. Insulin assays were established and performed at the Assay Centre, Department of Agriculture and Food, Melbourne University. Insulin levels were measured using a homologous double antibody radioimmunoassay (RIA) technique. The RIA was performed using purified insulin antiserum raised in guinea pig (Antibodies Australian) and purified bovine insulin for iodination and standards (Sigma-Aldrich, St. Louis, MI, USA, cat#I5500). Description of the full methods can be found in Ockenden et al. [18]. 

Commercial laboratories were used to analyse the immune biomarkers. The white cell count and differential were analysed using a Cell-Dyn 3700 autoanalyzer (Abbott Diagnostics, Abbott Park, IL, USA) at Gribbles Veterinary Pathology (Clayton, VIC, Australia). IBR antibodies titres were determined using a commercially available IBRGBC ELISA kit (IDvet, Grabels, France) at Veterinary Diagnostic Services-Agriculture Victoria (Bundoora, VIC, Australia). 

### 2.5. Statistical Analysis

The weekly body weight and physiological biomarkers were analysed using linear mixed models (LMMs) that were fitted using restricted maximum likelihood (REML). Fixed effects included in the LMMs were the effect of treatment and the effects of the covariates birth body weight and estimated BPI. The effect of calf was included as a random effect and was used as a residual term. This random effect was assumed to follow a normal distribution with zero mean and constant variance. Normality of distribution was examined using histograms of residuals and plots of residuals versus fitted values. The blood insulin measurements were logarithmically transformed to satisfy the assumption of normality with constant variance prior to the final analyses. The treatment means of blood insulin measurements were back transformed using the bias correction factor $\exp\left(\hat{\mu }+\frac{{\hat{\sigma }}^{2}}{2}\right)$, where $\hat{\mu }$ and ${\hat{\sigma }}^{2}$ is the estimated treatment mean and residual variance, respectively, in the logarithmic scale [22]. In all analyses, the individual calf was the unit of analyses. Any *p*-values that were less than 0.05 were considered statistically significant. All statistical analyses were performed using the statistical software GenStat 22nd Edition (VSN International, Hemel Hempstead, UK).

The number of calves per treatment was calculated based on the least significant difference formula, $LSD=t_{0.975}\sqrt{2{\hat{\sigma }}^{2}/n}$, $t_{0.975}$ being the 0.975 percentile of the *t* distribution and is approximately equal to 2, ${\hat{\sigma }}^{2}$ being the estimated variance in live weight from a similar previous experiment [23]. The *n*, number replication per treatment can be written as
$$n=2{\left(\frac{t_{0.975}\times \hat{\sigma }}{LSD}\right)}^{2}$$

We took 9.4 kg to be a reasonable *LSD* that we would like to detect between treatments in our experiment.

Three heifers from the Low group were removed from the herd at approximately 18 months of age (post immune challenge) when they failed to get in calf. 

## 3. Results

The 10 calves used for the experiment was chosen based on the sample size formula provided in the methods section.

At weaning, the heifers in the High treatment group were 19.0 kg (±2.3) heavier than the heifers in the Low treatment group (96.7 kg and 77.7 kg, respectively, *p*-value = <0.001) (Figure 1). The average daily gain (ADG) for heifers in the High group was 0.82 kg/day, compared with 0.89 kg/day for heifers in the Low group until the treatment groups converged at approximately 13 months of age. After this convergence, the ADG for both groups was approximately 0.7 kg/day. Seasonal fluctuations in growth rates are evident in both groups, particularly the High group from 8 months of age, where growth was slowed due to reduced feed quality post summer. Growth rates subsequently increase again during spring when pasture quality and quantity is high. 

Prior to vaccination, there were no significant differences in immune biomarkers between treatments (Table 1). However, 10-days post vaccination the WCC was significantly higher in the High treatment group (*p*-value = 0.038). Although not significant (*p*-value = 0.057), the neutrophil count in the High treatment group was approximately 30% greater than the Low treatment group (5.9 versus 4.1 × 10^9^/L). No significant differences were found between treatments in the remaining immune biomarkers measured post-vaccination.

There were no significant differences in any of the metabolic biomarkers tested between the High and Low treatment groups at 12 months of age.

There was no significant difference in the corrected skinfold increase post delayed-type hypersensitivity skinfold testing between treatments. Forty-eight hours post intradermal injections, the High treatment group measured a corrected skinfold increase of 6.0 mm, and the heifers in the Low treatment group measured a corrected skinfold increase of 5.8 mm when compared to the thickness prior to injection (SED = 0.55; *p*-value = 0.666).

## 4. Discussion

### 4.1. Growth Rates

Feeding a high volume of milk (twice the industry standard) in the preweaning phase did not result in advanced growth rates throughout early life. Calves in the High preweaning treatment group were 19.0 kg heavier at weaning than the Low treatment group calves; however, by 13 months of age this difference had disappeared. Treatment groups followed an almost parallel growth path immediately post weaning; however, after 6 months of age the Low group progressively caught up in bodyweight due to a slightly higher postweaning ADG (0.06 kg/day). It is common within calf growth studies that gains from accelerated milk feeding are lost postweaning [24,25,26] due to faster growth from the lower fed calves. Authors have attributed this growth pattern to the earlier and higher consumption of concentrates from restricted-milk-fed calves resulting in a less dramatic change in diet and a more developed rumen at weaning [25,26]. Alternatively, other authors have described this convergence of bodyweight as a consequence of a more severe growth check experienced during and immediately after weaning by accelerated-milk-fed calves. This period of slowed growth in accelerated-milk-fed calves allows the restricted-milk-fed calves to catch up in bodyweight [25,26]. 

Given the low concentrate intake reported in the preweaning phase by both treatments (<1 kg DM/calf/day) [18], there was surprisingly very little growth check experienced by either treatment group at weaning. Given the lack of difference in growth gains at this point, the low-milk-fed calves were unable to catch up in bodyweight at this time. As both treatment groups had low intakes during the preweaning phase, all calves would have undergone similar diet adjustments at weaning, and this was likely the reason for the similar growth rates [18]. Due to the faster growth that occurred in the subsequent months from the Low group, all heifers approached calving at similar bodyweights. Possible reasons for the faster growth rate from the Low group include the abundance of good quality pasture in spring or the occurrence of compensatory growth, which is common in ruminants [27]. On the contrary, Shamay et al. [28] found that restricted-milk-fed calves were able to compensate for skeletal size but not in bodyweight and their accelerated-milk-fed calves were still heavier at the point of calving. Further, also unlike some other studies [24,25,26], Shamay et al. [28] found their accelerated-milk-fed calves had greater milk yields in their first lactation. A possible reason for these differing results could be due to differences in methodology. As well as variations in milk volume, Shamay et al. [28] also included a variation in milk type (whole milk vs. milk replacer); therefore, it cannot be determined if it was the preweaning nutrition (amount and/or type of milk) or bodyweight at calving that influenced the difference in milk yield in their experiment. The benefit of having all heifers enter their first lactation at similar bodyweights in the current experiment is the elimination of bodyweight as a confounding factor on milk yield; therefore, any differences detected would be a direct result of the preweaning nutrition and subsequent development. Further research is warranted to determine what effect preweaning nutrition has on the future growth patterns and the physiological development of these heifers and, additionally, what effect these have on the subsequent milk yield. 

### 4.2. Immune Biomarkers

Calves from the High treatment group maintained superior immune characteristics at 12 months, following a period of 9 months on a common diet. The higher WCC (*p*-value = 0.038) and neutrophil count (*p*-value = 0.057) in the High group calves at 12 months of age mimicked what was seen during the first immune challenge at 6 weeks of age when the nutritional treatments were being applied [18]. The preservation of these trends may suggest evidence of a form of immunological developmental programming or imprinting in the preweaning phase. Somewhat similar findings were reported by Aleri et al. [19]. Whereas the authors did not describe the preweaning nutritional circumstances, immune responsiveness was preserved within heifers tested at 5–6 months of age and then again at 12–13 months of age. However, unlike the current experiment, it was also reported that higher immune responsiveness coincided with a higher ADG. The immune challenge described in the current paper commenced at 12 months of age. During this month, ADG was still numerically higher for the Low treatment group and differences in bodyweights between treatments had disappeared. It would therefore seem plausible that preweaning nutrition, not current bodyweight or ADG, is accountable for the results obtained in the current immune challenge. 

### 4.3. Metabolic Biomarkers

The concentrations of metabolic biomarkers including BHB, NEFA, glucose and insulin were the same for the High and Low treatment groups. In the preweaning phase, calves in the Low treatment group were in a poorer metabolic state pre-vaccination, which was exacerbated post vaccination [18]. Given the lack of difference in metabolic biomarkers in the current experiment, it is reasonable to believe that unlike the immune biomarkers, the metabolic findings in both experiments were a direct result of the nutritional input at the time. However, as previously discussed, the High treatment group did generate a superior immune response to the vaccine, with no effect on their metabolic biomarkers. Therefore, it seems this superior immune response did not come at a metabolic cost or had no detrimental effects to other processes such as growth or, potentially, future lactations. When interpreting the lack of preserved differences in metabolic biomarkers alone, these trends are supported by Rodríguez-Sánchez et al. [29] and De Paula et al. [30], who also reported that metabolic differences detected in the preweaning phase were not detectable later in life. However, Kenéz et al. [15] found long-term differences in the metabolic patterns of calves fed ad libitum or restricted levels of milk in the preweaning phase. Further studies with greater animal numbers are warranted to fully investigate the influence of early life nutrition on physiological programming. 

### 4.4. Delayed-Type Hypersensitivity Skinfold Testing

There was no difference in the delayed-type hypersensitivity skinfold testing reactions between treatments. This result was expected as there were also no differences observed in the preweaning phase [18]. Given bodyweight and ADG were similar for both groups, this is further supported by other studies such as Aleri et al. [19] and Foote et al. [31] who also found no relationship between ADG and skin welt increase post delayed-type hypersensitivity skinfold testing.

### 4.5. Reproductive Performance 

Despite the small sample size, it would be incomplete not to mention the reproduction outcomes of these heifers. Three of the ten heifers in the Low treatment group did not get in calf, whereas all the High treatment group did. Given the Low treatment group were subject to the current industry standard of preweaning nutrition, these results support claims that approximately 30% of heifers do not reach their second calving [2]. It would therefore seem suitable to investigate the effect preweaning nutrition has on the reproductive performance of dairy replacement heifers. 

## 5. Conclusions 

Results from this series of experiments indicate that calves with a high milk intake in the preweaning phase have superior immune characteristics when compared with low-milk-fed calves, not only at the time of increased nutrition but until at least 12 months of age. Given both groups were under non-experimental conditions postweaning, the results from the current paper may suggest evidence of a form of immunological developmental programming. To produce calves with superior immune responses would contribute to the generation of more resilient cows. More resilient cows are physiologically better equipped to handle immune challenges and would be less likely to have their milk production compromised due to pathogenic or metabolic disease. However, this experiment is limited by sample size and more work with larger numbers is required to further validate these findings and to determine the effect preweaning nutrition will have on the first lactation of these heifers. 

## Figures and Tables

**Figure 1 animals-13-01309-f001:**
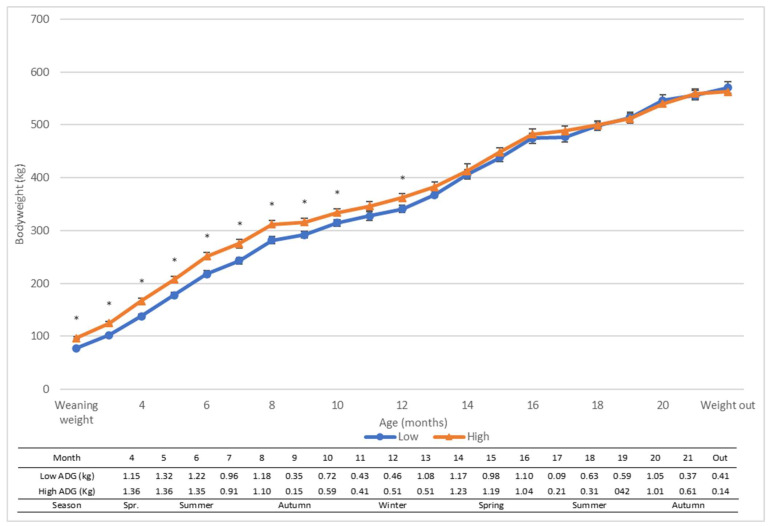
Body weight (kg) and average daily gain (ADG) (kg) of weaned heifers offered 4 L (Low) or 8 L (High) of milk daily in the preweaning phase before no differential treatment postweaning. Asterisks * indicate significant differences (*p*-value < 0.05). Error bars depict the standard error of the mean (SEM).

**Table 1 animals-13-01309-t001:** Immune and metabolic biomarkers pre-vaccination and 10 days post-vaccination of 12-month-old heifers offered 4 L (Low) vs. 8 L (High) of milk daily in the preweaning phase and no differential treatment post weaning (WCC, white cell count; IBR, infectious bovine rhinotracheitis; BHB, beta-hydroxybutyrate; NEFA, non-esterified fatty acid).

Time	Biomarker	Low	High	SED	*p*-Value
Pre-vaccination					
	WCC ^1^	7.6	8.8	0.94	0.207
	Neutrophils ^1^	2.5	3.6	0.87	0.246
	Lymphocytes ^1^	4.4	4.6	0.48	0.767
	Monocytes ^1^	0.4	0.4	0.10	0.754
	Eosinophils ^1^	0.2	0.3	0.09	0.482
	Basophils ^1^	0.0	0.0	-	-
	IBR Titre (S/N%)	37.6	24.4	14.65	0.381
	BHB ^2^	0.29	0.23	0.039	0.163
	NEFA ^2^	0.36	0.35	0.037	0.734
	Glucose ^2^	4.62	4.61	0.186	0.968
	Insulin (µIU/mL)	12.32	11.18	-	0.762
	Logₑ Insulin	2.27	2.17	0.316	
10 days post- vaccination					
	WCC ^1^	8.8	10.9	0.94	0.038 *
	Neutrophils ^1^	4.1	5.9	0.91	0.057
	Lymphocytes ^1^	3.7	3.5	0.49	0.660
	Monocytes ^1^	0.6	0.8	0.15	0.210
	Eosinophils ^1^	0.3	0.6	0.18	0.130
	Basophils ^1^	0.0	0.0	0.02	0.654
	IBR Titre (S/N%)	5.2	6.4	2.49	0.651
	BHB ^2^	0.32	0.31	0.045	0.890
	NEFA ^2^	0.29	0.26	0.028	0.274
	Glucose ^2^	4.98	4.81	0155	0.296
	Insulin (µIU/mL)	18.11	13.22	-	0.269
	Logₑ Insulin	2.71	2.40	0.274	

^1^ Values are ×10^9^/L; ^2^ Values are in mmol/L; Asterisks * indicate significant differences (*p*-value < 0.05).

## Data Availability

The data available in this study are available on request from the corresponding author. The data are not publicly available due to the experiment was conducted within Agriculture Victoria and are therefore bound by their policies.

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
