# Peer review of "The Preservation of the Effects of Preweaning Nutrition on Growth, Immune Competence and Metabolic Characteristics of the Developing Heifer"

_animals, 2023, doi:10.3390/ani13081309_

Round 1
Reviewer 1 Report
Please refer to the file uploaded.
Read carefully the manuscript and make the corrections.

Reviewer 2 Report
In the manuscript ID animals-2297080, the authors investigated the effects of milk feeding levels during preweaning on dairy heifer in Australia on feeding performance and immunity. This experimental data is useful information for the Australian and global dairy cattle industry. However, this manuscript needs some improvements.
1) Authors did not clearly indicate the scientific novelty of the study in the introduction. Are there any previous papers examining the effects of accelerated nutrition during the preweaning phase on post-weaning heifer immunity?
2) Authors should indicate the range of P-values that determine significant differences in 2.5 statistical analyses.
L214-216 Add the result of ADG in Table.
L216-224 Include seasonal information on Figure1.
L241-245 Add the results of the skin fold test in Table or Figure.
L330-337 I recommend showing reproductive performance in Results and Table.
Reviewer 3 Report
Overall, this is a much needed topic for the industry. These type of data require a long time to collect, so I commend the authors for evaluating. Article is well written. Below are my comments wanting some more information about the enrollment procedure as well as more description about the treatment groups.
Sample size calculation-Was a sample size done before the study, and if so please provide. Need some justification of the small sample size used for the study and it's potential limitations.
2.1. Please include frequency of feedings as well as milk used (milk replacer or waste milk or whole milk). Was the 4 L group given an additional 4 L of water through a bottle, so total daily volume and feeding frequency the same between treatment groups? If not, need to include as limitation of the study as the initial weight gained could have been due to just increased volume and increased feeding frequency. Were calves housed in individual hutches.
How many days did it take to enroll calves? If multiple days, was the immune challenge provide to calves on the same day or same day of age?
Was any measurement of colostrum transfer measured on the calves? If so, please include and provide results to make sure treatment groups were not different from the beginning. Would be good to include colostrum management in the description as well.
Figure 1-Are those supposed to be error bars in the graph? The lines and bars don't match up.
Round 2
Reviewer 3 Report
Please add in some detail of sample size calculation in the manuscript, as I believe that is an important concept.
I do not believe LSD are appropriate because only 2 treatment groups. You already have marked which ones were significant based on *. Providing standard errors for each treatment group might be beneficial though. I will defer to the editor for final decision.
